# E. S. Fedorov Promoting the Russian-German Scientific Interrelationship †

**Peter Paufler** [1,*] and **Stanislav K. Filatov** [2]

1   Inst. f. Festkörper- u. Materialphysik, Fakultät Physik, TU Dresden, 01062 Dresden, Germany
2   Department of Crystallography, St. Petersburg State University, 199034 St. Petersburg, Russia; filatov.stanislav@gmail.com
*   Correspondence: peter.paufler@t-online.de; Tel.: +49-(0)351-4210411
†   In parts presented by the corresponding author during the XIX International Meeting on Crystal Chemistry, X-ray Diffraction and Spectroscopy of Minerals, Apatity, Russia, 2–5 July 2019.

**Abstract:** At the dawn of crystal structure analysis, the close personal contact between researchers in Russia and Germany, well documented in the "Zeitschrift für Krystallographie und Mineralogie", contributed significantly to the evolution of our present knowledge of the crystalline state. The impact of the Russian crystallographer E. S. Fedorov upon German scientists such as A. Schoenflies and P. Groth and the effect of these contacts for Fedorov are highlighted hundred years after the death of the latter. A creative exchange of ideas paved the way for the analysis of crystal structures with the aid of X-ray diffraction.

**Keywords:** E. S. Fedorov; A. Schoenflies; W. Barlow; space group types; Russian-German interrelations; Zeitschrift für Krystallographie und Mineralogie

---

In commemoration of the 100th anniversary of the death of Evgraf S. Fedorov.

## 1. Introduction

The eminent Russian crystallographer Evgraf Stepanovič Fedorov (1853–1919) (Figure 1) has received numerous tokens of appreciation for his pioneering contributions to the fundamentals of crystallography (e.g., [1–15]). His most important contributions, the derivation of 230 space group types and the methods of the experimental survey of crystal morphology, are named after him. Since he drew attention to the great importance of apatite resources for agriculture early on, the city Apatity (Kola peninsula) was an appropriate location to acknowledge his work.

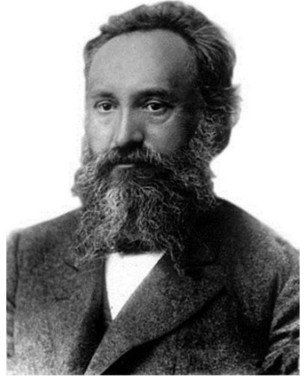

**Figure 1.** Evgraf Stepanovič Fedorov (https://de.wikipedia.org/wiki/Jewgraf_Stepanowitsch_Fjodorow).

On the occasion of the 100th anniversary of his death, we will briefly pay attention to one aspect of his oeuvre, relating the scientific communities in Russia and Germany. When quoting German words, the spelling of the original document is used. Hence, 'crystallography' is spelled in German 'Krystallographie' before the German spelling reform in 1921 and 'Kristallographie' afterwards. Readers accustomed to modern German should be aware of that, because it applies to some more terms in the text or in the references. We refer to 72 publications of E. S. Fedorov (see Table S1) in the 'Zeitschrift fuer Krystallographie und Mineralogie' (founded in 1877; abbrev. "Z. Kryst.") and focus on their impact upon crystallography. The latter journal played a leading role in crystallography during the turn of the 19th century Fedorov's papers on philosophical matter will thus be disregarded here [9,12]. Z. Kryst played a leading role in crystallography during the turn of the 19th century. Thus, his results became known worldwide with special influence upon science in German-speaking countries. On the other hand, due to the language barrier, his publications in German will nowadays not find due reference as previously; the present contribution may help to improve this. In the second half of the 20th century Z. Kryst. switched over to English ((Zeitschrift Kristallographie—Crystalline Materials) and additional journals, specialized in crystallography, came up, e.g., Acta Crystallographica; Кристаллография(English translation: Crystallography Reports); Crystal Research and Technology).

A salient point of that was the fruitful interaction between the author Evgraf Fedorov and the German Editor of Z. Kryst., Paul Groth [16], documented by Šafranovskij et al. [7]. Letters have been exchanged during the period 6/18 October 1891 till 5/18 July 1914. This contact offered Fedorov the opportunity to publish results which indicated the wide scope of his interests, whereas Groth profited a lot from the high level of submitted articles for the reputation of the journal. We will structure the following along major topics dealt with by Fedorov in Z. Kryst.

Before going into detail, we want to stress the remarkable fact that Fedorov and his direct male descendants form a four-generation dynasty of geologists: Evgraf Stepanovič Fedorov (1853–1919), Evgraf Evgrafovič Fedorov (1880–1965), Èlij Evgrafovič Fedorov (Data of birth and death not available) and Vadim Èl'evič Cubin (Fedorov) (1940–present). Vadim Èl'evič's mother decided to give him her family name 'Cubin'. According to the own words of Vadim Èl'evič [17], this geological line breaks off, although there have been two more generations of offspring: his children and grandchildren. One of the authors (S. K. Filatov) performed a study at the Geological Faculty of the St. Petersburg State University (LGU-SPbGU) in the group of geochemists together with Fedorov's great-grandson Vadim Cubin: Vadim at the Department of Petrography followed his great-grandfather as well as his mother and S.F. at the Department of Crystallography. Vadim was proud of his great-grandfather, however, consistent with his modesty, his fellow students did not learn about his ingenious great-grandfather before graduation, when Vadim Èl'evič started a job with VSEGEI (Vsesojuznyj Geologičeskij Institut) and his friends unveiled the secret of his renowned ancestor. Fifty years after leaving the university, he visited the Department of Crystallography of SPbGU, where a photograph was taken (Figure 2).

While the scientific activity of E. S. Fedorov will be elucidated by the following chapters, reference to the life of scientists and the atmosphere around the intelligentsia in Russia at the end of the 19th and the beginning of the 20th century will be given only briefly. Fortunately, vivacious reminiscences have been left by Fedorov's wife Ludmila Vasil'evna Fedorova, née Panjutina, pupil of the 'Smol'nyj Institut' and gifted writer [18]. She became a strong support of her husband in scientific and everyday life against the background of the dramatic Russian history. Three children (one son, two daughters) belonged to the family.

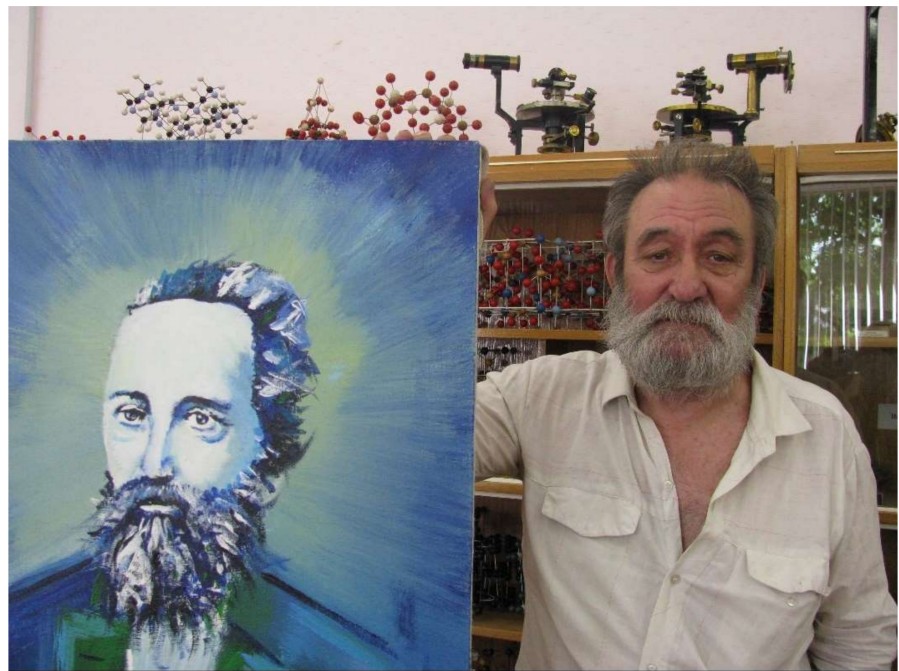

**Figure 2.** Vadim Èl'evič Cubin (Fedorov) (**right**), great-grandson of Evgraf Stepanovič Fedorov, when he visited the Department of Crystallography of SPbGU—accompanied by his wife Ljudmila Vasil'evna—50 years after finishing his studies on the occasion of a festive session of the Department. He shows the portrait of his great-grandfather, done by the young artist Roman Kajukov—student at the Department. Middlesized, chunky and with full beard Vadim shows his resemblance to his famous ancestor (Photo by S. K. Filatov).

## 2. On the Congenial Action of E. S. Fedorov and A. Schoenflies

There is no doubt that the derivation of symmetry groups for crystals (230 space group types or Fedorov Groups) remains the most important scientific achievement of E. S. Fedorov (Figure 3) (230 space-group types designate the number of combinations of symmetry operations under the condition of three-dimensional translation symmetry. Space-group types are often termed 'space-groups'. Strictly speaking, the magnitude of the translation vectors has to be taken into account, when assessing the total symmetry. While the term 'space-group' includes specific translations, the 'space-group type' is independent of them). The history of this discovery is a striking example of how natural science works and has been appreciated by a number of authors [1–4,6–15]. We will put the role of the German mathematician Arthur Schoenflies on record, who utilized group theory to work out Sohncke's (Leonhard Sohncke (* 1842, † 1897), German physicist and crystallographer at Karlsruhe, Jena and Munich) classification of regular point systems [19] and, encouraged by Felix Klein ((* 1849, † 1925), German mathematician at the University of Göttingen), proceeded with the inclusion of improper movements of space [20–24]. For details of his course of life see Reference [10].

In the regular meeting of the Imperial Mineralogical Society S. Petersburg on 21 November 1889 (Julian or Old Style calendar) [25], Fedorov announced the completion of his work "Simmetrija bezkonečnych pravil'nych system figur" ("The symmetry of infinite regular systems of figures"). The final version is displayed in Figure 3. Moreover, he informed about the accidental taking notice of a publication of Schoenflies [21], dealing with the same subject. Fedorov had noted that Schoenflies enumerated 227 symmetry groups, one less than he did. Schoenflies, in turn, mentioned Fedorov for the first time in [24].

**Figure 3.** First page of E. S. Fedorov's publication "Symmetry of regular systems of figures", which appeared in *Trans. Mineral. Society St. Petersburg*, 1891 [26].

At the end of 1890 Fedorov submitted a summary of his works on symmetry in German to the "Neues Jahrbuch für Mineralogie, Geologie und Paläontologie" [27]. Between 1889 and 1908, Schoenflies and Fedorov started a fruitful correspondence [4,6,28] as part of the interrelationship addressed in the title of this paper.

In his first letter dated 14 December 1889, Schoenflies sent his papers published in 1888 [22,23] to Fedorov with the statement "Über die Übereinstimmung mit Ihren eigenen Anschauungen spreche ich meine große Freude aus … Die Priorität gebe ich Ihnen gern zu … " [4] (p. 92) ("Happy to see the convergence of our joint ideas. … I gladly recognize your priority … " (see also Reference [14], p. 394).

Fedorov concluded in 1892, comparing his and Schoenflies' results [29] (p. 26): " … nachdem wir Beide die ersten Resultate dieser Thätigkeit publicirt haben, tritt eine höchst wunderbare Thatsache zu Tage: eine solche Uebereinstimmung in der Arbeit zweier Forscher, wie vielleicht die Geschichte der Wissenschaft kein anderes Beispiel aufzustellen vermöchte" ("Having separately published the first results of this activity, an extreme wonderful fact emerges: for such an agreement of the work of two researchers, the history of science might not have in store another example").

Moreover, he underlined the mutual influence of a Russian and a German research project by pointing out " … dass das Wesentlichste, was den zweiten Abschnitt des neuen Buches von Schoenflies ausmacht (über 400 Seiten umfassend) (See Reference [30] and Figure 4), sich als unsere gemeinsamen Resultate angeben lässt, welche zum grössten Theile bisher noch von keinem Forscher publicirt wurden." (" … that the essentials, which constitute the second chapter of Schoenflies' new book (comprising more than 400 pages), may be considered our common results, which have not yet been published elsewhere by any researcher").

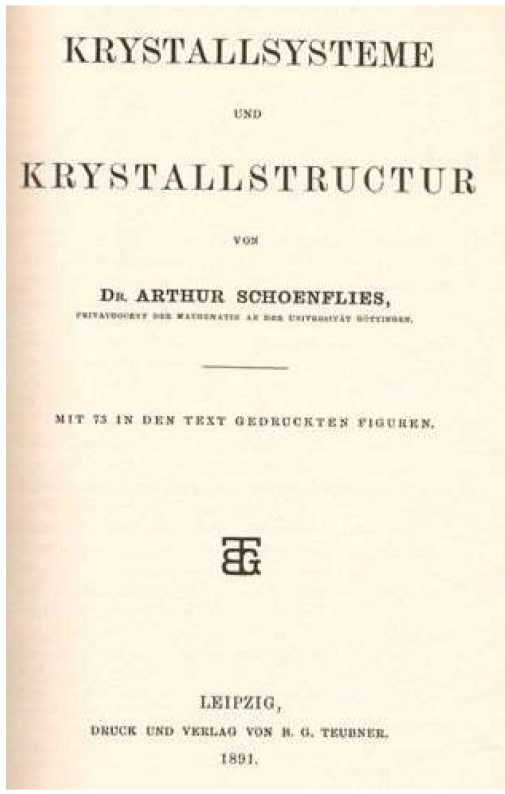

**Figure 4.** Title page of A. Schoenflies' book "*Crystal Systems and Crystal Structure*", published by B.G. Teubner, Leipzig 1891 [30].

Schoenflies praised Fedorov's effort [31] to present that direct comparison, which he himself had in mind to publish. Additionally, he referred to a few differences in the definition of terms. The friendly relationship between Fedorov and Schoenflies did not prevent them from serious criticism against each other (cf., e.g., [4,25]) (Fedorov complained about Schoenflies' account of the chronology of their joint findings [32], p. 588). Further correspondence was devoted to special mathematical questions, in which also F. Klein has been involved [4,6,29]. Then, communication broke down until 1903 (The final correspondence is dated 14 May 1908 in [4]).

## 3. Evgraf von Fedorov and Paul von Groth Promoted Significant Advances in Crystallography

### 3.1. Getting E. S. Fedorov as Author, Referee and Consultant

Fedorov's first paper in Z. Kryst. [29] was the result of more than 3 years of discussion between him and Schoenflies and contained the first correct list of the 230 space groups in a journal. Apart from the beneficial feature of this article to facilitate transition from the nomenclature of Fedorov to that of Schoenflies, Sohncke or Hessel (Johann Friedrich Christian Hessel (* 1796, † 1872), German crystallographer, Professor in Marburg), Fedorov's expressions to characterize symmetry employing tools of analytical geometry were given. They proved an efficient vehicle to get along with symmetry operations. For a more recent comparison of various notations employed for space groups see, for instance, Hilton [33], Nowacki [34] and Paufler [35].

From this time on Evgraf von Fedorov started an extensive correspondence with the founder of Z. Kryst., Paul Heinrich Groth (Figure 5) [16,36–39], as of 1902 Ritter von Groth (* 1843 Magdeburg; † 1927 Munich). Groth was Professor of Mineralogy at the University of Munich from 1883 till 1924 and, at the turn of the century, the leading German crystallographer. This contact remained during the entire period 1891–1914 before World War I. It was more than businesslike exchange of letters.

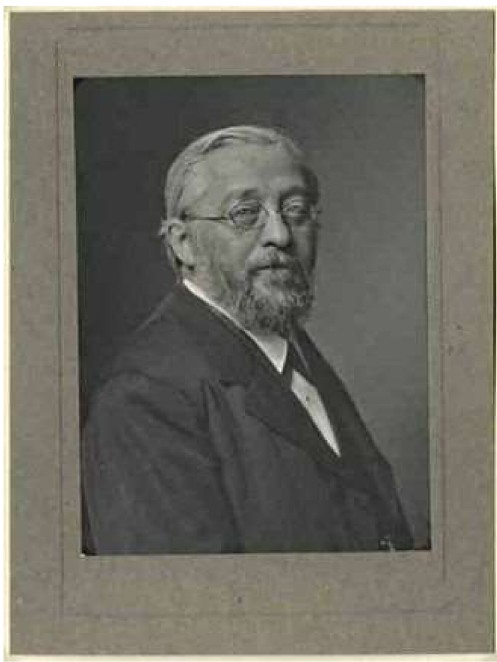

**Figure 5.** Paul Heinrich von Groth (https://de.wikipedia.org/wiki/Paul_Heinrich_von_Groth).

Fedorov signed his letters to Groth differently: till 1896 mostly "E. Fedorow"; 1896–1898 "Ew. Fedorow"; 1898–1906 "E. von Fedorow"; 1906–1914 "E. S. Fedorow". As designated member of the Bavarian Academy, Fedorov provided some personal data. He specified his profession as Mining Engineer and Professor of the Moscow Agricultural Institute and his descent as from a Russian noble family [7] (p. 72), hence, the nobiliary prefix "von". His father was major-general, i.e., according to the rules Fedorov was raised to the personal peerage [10]. To give a signal of his disappointment at the failure of the Russian nobility during the revolution in 1905, he announced in a letter to Groth, dated 15/28 November 1906, that he will formally renounce the social affiliation to nobility, i.e., skip "von" when signing letters. Publications in Z. Kryst., however, continued to appear under the authorship 'E. von Fedorow'.

The correspondence showed quite nicely the response of both scientists to actual new developments and contributed a lot to the mutual understanding. Groth also played an active role in paving the way for Laue's discovery of X-ray diffraction by crystals [16,36–39].

As Corresponding Member of the Russian Academy of Sciences St. Petersburg since 1883, Groth was well known among Russian mineralogists and familiar with the Russian Mineralogical Society. He was also aware of the rejection experienced by Fedorov in the Academy. Unaffected by personal hostilities against Fedorov, Groth highly estimated the scientific achievements of the latter and nominated him, together with Tschermak (Gustav Tschermak (* 1836 Littau, † 1927 Vienna), Austrian Mineralogist), for the Award of the St. Petersburg Mineralogical Society, which he received on 7 December 1893 ([7], p. 287) in appreciation of the theodolith method [32].

Moreover, in the same year, Groth nominated Fedorov for election to the Academy, although without success [7] (p. 45). Hence, he decided to by-pass those internal obstacles and nominated E. S. Fedorov, together with L. Sohncke, for election as Corresponding Member to the Bavarian Academy of Sciences, Munich. When Fedorov on 19 November 1896, in appreciation of his extraordinary scientific achievements, received the diploma, and thus, more international recognition, his professional situation had been stabilized already since 1895 by being appointed Full Professor of Geology. Fedorov noted in a letter to Groth, that in Russia he is not considered crystallographer. However, he needed the job and had accepted ([7]; p. 57) the appointment at the Moscow Agricultural Institute. Expressing his sincerest thanks for the honorable election, he attached particular importance to the fact, that a German

Academy had elected him. He held that they never were guided by political attitudes but rather decide following strict scientific and objective arguments [7] (p. 72).

Between 1891 and 1914, many letters have been exchanged between Fedorov and Groth, at least 223 of them were preserved [7,39]. Furthermore, as mentioned above, Fedorov published 72 papers in German (Table S1). He had at least basic knowledge of the German language. For the copy of a handwritten letter see Reference [7] (p. 21). This may be because he attended the German 'Annenschule' St. Petersburg (1863–1867) [39]. Moreover, as politically active student [12], he visited Germany in 1877, where he met W. Liebknecht and A. Bebel [10]. Groth offered him support with stylistic improvements of manuscripts in German [7] (p. 21), which Fedorov explicitly asked for [7] (p. 74). Obviously, Groth, in turn, was helped by co-workers in Munich, e.g., by Dr. F. Hofmann [32] (p. 679). Note that Groth asked Fedorov to recommend somebody, who would be willing to report in German language on articles written in Russian for Z. Kryst./section 'Extracts'. He named in particular G. Wulff (Warsaw, from 1908 on: Moscow) for that, who then supplied several abstracts of Fedorov's publications in German. Wulff seemed not to be involved in the translation of Fedorov's original contributions to Z. Kryst. Several articles published by Fedorov in the Russian 'Zapiski imperator. S. Peterb. Mineralog. Obščestva' have been extracted in German by Th. V. Barker. Nowadays, it is rare to find editors taking personal interest in both the science and personality of authors on the one hand and authors contacting editors on a personal level on the other.

After the election to the Bavarian Academy, Groth advised Fedorov to pay a visit to members of the Academy in order to become acquainted with them and other scientists in Germany. In particular, Groth requested the acquaintance of Fedorov that way to learn more about the Universal Method [32], not only as editor of Z. Kryst., but also as author of a textbook [40]. Fedorov chose the Christmas Holidays 1898/1899 to visit Berlin, Heidelberg, Vienna and Munich and met several representatives of Mineralogy there. He felt that the meeting with Groth in Munich differed from most of the other contacts due to special heartiness [7] (p. 190). The contact with the President of the German Mineralogical Society, Klein (Johann Friedrich Carl Klein (* 1842, † 1907), Professor of Geology, University of Berlin), shall be mentioned here, which sheds light on the character of Fedorov. Klein offered his services to him to do public relations work via the German Government directly to the Russian Tsar in order to push Fedorov's career. Fedorov refused this kind of protection stating that a veritable scientist should not approach higher governmental authorities to pursue interests of his own [7] (p. 189).

Both Groth and Fedorov were teaching crystallography, the former in Munich and the latter in Moscow/St. Petersburg. Being well-connected, they exchanged teaching aids. For example, Groth made use of Fedorov's Textbook of Crystallography and of instruments developed by Fedorov for lectures and practicals. On the other hand, Groth supplied his Textbook of Physical Crystallography to him.

Groth and Fedorov agreed in seeking precise terms of crystallographic content, which is reflected by an extensive debate on designations [7] (pp. 27,34). From the very beginning of his studies in the field of crystallography and mineralogy, Fedorov felt that progress in this rapidly growing discipline will require suitable and clear concepts and notions. One of his essential contributions to the advancement was his permanent criticism of inconsistent designations utilized by other authors (e.g., Mohs [32], p. 579; Groth [41], p. 221; Naumann [32], p. 577) or his arguing against errors in publications (For example, Fedorov not only criticized an error in a calculation published by Theodor Liebisch, he also complained about the disregard of his criticism in ensuing publications [32], p. 700). With suggestions for standardized designations in "ZKryst" he became known to the international community (He demanded: " . . . Die Fachwörter sollen ein einheitliches System bilden." (" . . . Technical terms must form a consistent system . . . ") [32], p. 578) (Figure 6).

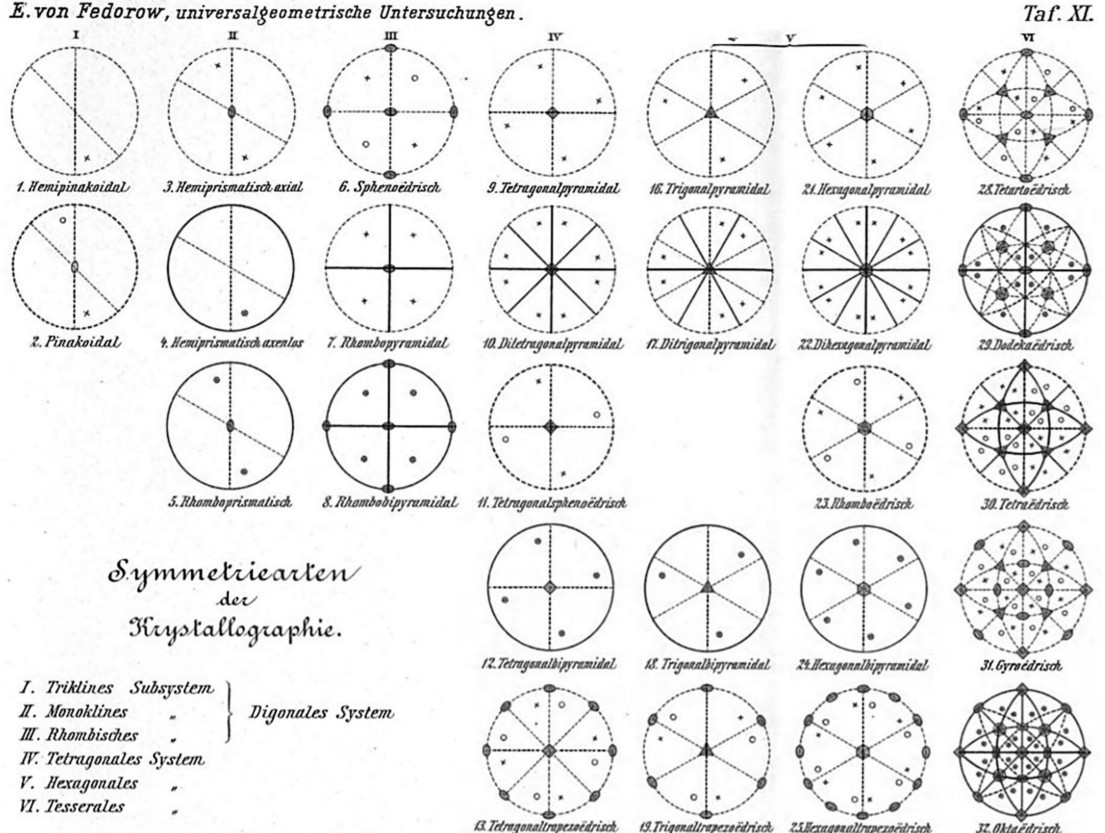

**Figure 6.** Part of the figure summarizing symmetry types "Symmetriearten der Krystallographie" which was published by Fedorov in Z. Kryst. 1893 [32] (Table XI). Note that standardized graphics of stereographic projections of symmetry axes and planes, poles of general position as well as designations of each class below the graphics are given (Fedorov-Groth nomenclature).

On the basis of new insight into the symmetry properties of regular point systems, Fedorov's ambition to standardize designations and symbols in the field of crystallography found additional objects. In 1895 he published graphical representations of symmetry elements for each of the 230 space group types [41] and in the following year parallelohedra consistent with them [42]. According to the state of the knowledge, he supposed that crystal morphology reflected internal structure. Fedorov attributed a system of designations to the parallelohedra, which was the result of hard discussion with Paul Groth.

In an appendix to that paper [42], Fedorov made a positive statement with respect to the degree of agreement: "Es gewährt mir eine sehr grosse Genugthuung, in dem eben erschienenen Buche "Physikalische Krystallographie" [40] von einem so namhaften Specialisten, wie Herr Prof. P. Groth, die Annahme einer krystallographischen Nomenklatur zu sehen, welche mit der von mir seit Jahren in allen meinen Werken gebrauchten wesentlich übereinstimmt" ("I am very much gratified to see a crystallographic nomenclature being accepted in the book "*Physical Crystallography*"—published quite recently by such renowned specialist as Prof. P. Groth—which is in substantial agreement with the one I am using for years.") Some disagreement, however, remained.

*3.2. New Experimental Tools for Crystallographic Studies*

The second column of Fedorov's work comprises the development of the universal goniometer (Figure 7), the universal table for optical microscopy (Figure 8) and various new methods for evaluation of the data obtained herewith. Groth took a special interest in that and stimulated Fedorov to submit original results (Examples are Nos. 2, 4, 7, 9, 12, 13, 15, 17–19, 25, 26, 34, 35, 39, 45, 60, 67, 68 in Table S1)

and translations of previously published Russian articles (For example Nos. 5 and 6 in Table S1) to Z. Kryst. Furthermore, extensive German excerpts of important Russian papers of Fedorov were published by that journal (Among them examples highlighting experimental achievements, which facilitate daily work, e.g. [43–49]). A report on Fedorov's important two-circle goniometer or theodolith goniometer appeared in 1893 [32] almost simultaneously with Goldschmidt's [50] and Czapski's [51,52] variants of a very similar device. The priority, however, is due to Fedorov, because of a brief note presented already in 1889 at the Mineralogical Society St. Petersburg (cf. [32], p. 574).

At least comparable attention attracted the universal stage (U stage or Fedorov table, Figure 8) developed by Fedorov to measure optical properties of crystals. Up to five optical values can be measured, depending on the symmetry of the crystal: three principal indices of refraction, the optic axial angle and the orientation of the principal vibration directions with respect to the crystallographic coordinates [53]. High precision of mechanical work is required, and a great deal depends on the skill of the mechanic. Fedorov mentioned thankfully the work of mechanic Petermann (cf., e.g., [32], p. 603). Groth realized the potential demand of this equipment and arranged the contact to the German mechanic Heinrich Ludwig Rudolf Fuess (* 1838, † 1917), who was the leading manufacturer of optical instruments like this in Germany. Stimulated by Fedorov's publication, Fuess made a specimen, which induced Fedorov to consider the pros and cons [7] (p. 61), this way giving rise to further improvement.

Dealing with data obtained with the aid of the theodolith goniometer and Fedorov's table Fedorov needed tools to handle the huge mass of stereographic projections (Figure 9). Therefore, in 1896 he proposed to print stereographic nets (10 cm in diameter) to facilitate stereographic projections of crystals by every crystallographer [7] (pp. 73,74). Later on, the printing-office Wilhelm Engelmann (Leipzig, Germany) offered samples to do that. Even during the 20th century, many textbooks on crystallography enclosed a standard projection of great and small circles for the reader's convenience.

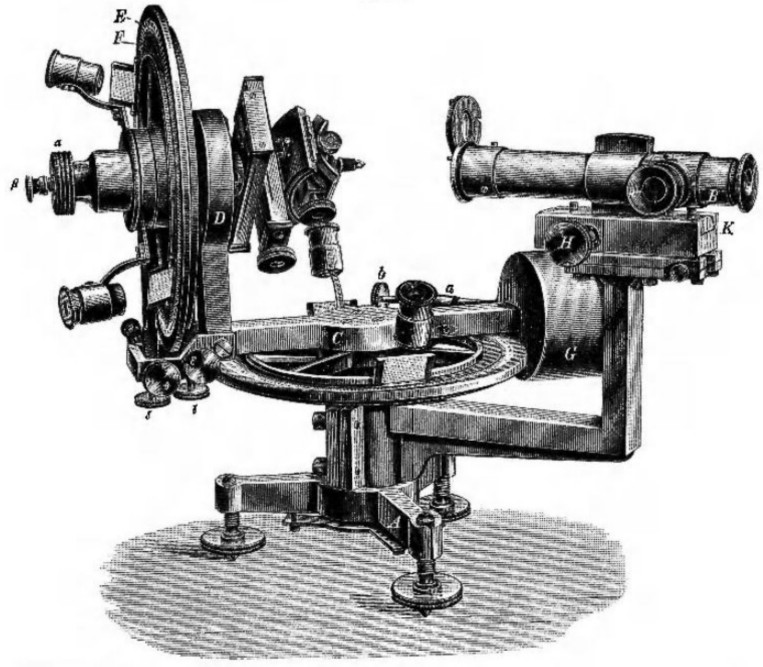

*) Die erste Beschreibung des Apparates und die Angaben der mittelst derselben erzielten Vortheile publicirte ich schon i. J. 1889 (Verhandl. d. k. Mineral. Ges. **26**, 458). Neuerdings hat der Verfertiger desselben, Herr Mechan. P e t e r m a n n, für die kais. Akad. d. Wiss. ein zweites, mehrfach verbessertes Exemplar hergestellt.

**Figure 7.** Universal (Theodolite) Method in Mineralogy and Petrography. In the footnote Fedorov mentions that the first version of this equipment has been published already in 1889 and that the mechanic Petermann recently performed a second sample, which has been improved several times. (E. v. Fedorov [32], p. 603, Figure 3).

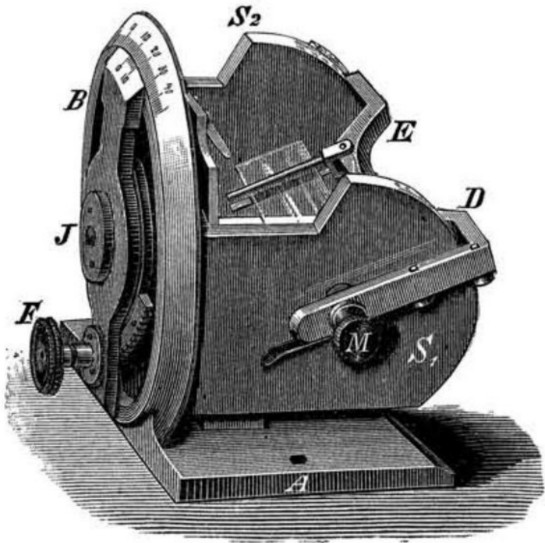

**Figure 8.** Microscope Universal Stage (U-Stage or Fedorov stage) with independently tilting (*J*) and rotating (*M*) axes. The specimen is placed on a glass plate (two points of the sample holder D and E are indicated, which may be rotated around axis M). The two axes enable to orient crystallographic and optical planes of the crystal to the reference system of the microscope. The device allowed quick measurements of optical properties using thin sections of minerals. Later on, it has been refined with five axes of rotation for the study in any cross section. (E. v. Fedorov, [53], p. 235, Figure 2).

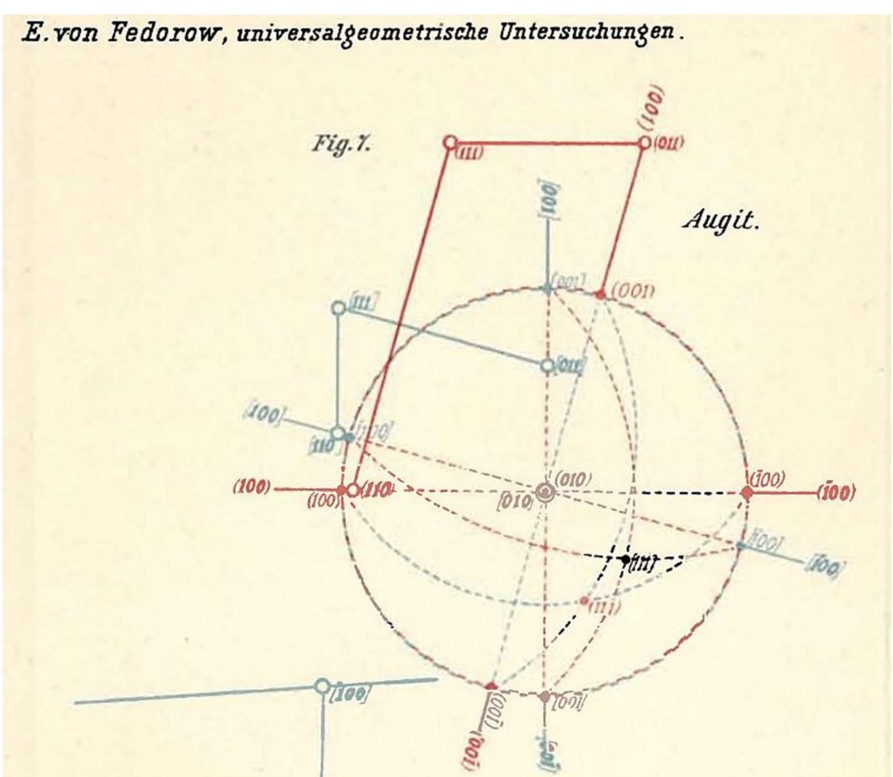

**Figure 9.** Universal geometrical studies. Combined stereographic and gnomonic projection of crystal forms and edges used by E. v. Fedorov. Faces appear in red color, zones in blue [32] (Table XIII).



### 3.3. From Morphology to Crystal Chemistry

The availability of space groups very much stimulated structure proposals at the end of the 19th century (e.g., [54]). Since Fedorov's pioneering work, numerous experimental data of crystal structures have been collected in databases. Examples are the "Inorganic Crystal Structure Database (ICSD)" containing in 2019 more than 200.000 inorganic crystal structures and the "Cambridge Structural Database (CSD)," which at the end of 2019 reached one million small-molecule organic and metal-organic crystal structures.

Although Fedorov found in total 230 symmetries allowed for structures in three-dimensional space, it soon became clear that their abundance is not evenly distributed among the space groups. Certain combinations of space symmetry are obviously preferred for the packing of given structural units. While the most frequently observed space group of minerals and inorganic compounds is the orthorhombic *Pnma* (No.62, see the white background of the Graphical Abstract), the tetragonal space group $P4_2cm$ (No.101, black background) thus far seems to be avoided completely by a mineral (Filatov [55], see also Graphical Abstract). The distribution depends, however, on the symmetry of the structural unit. Organic compounds, for instance, occur most frequently with a crystal structure of the monoclinic space group $P2_1/c$ (No. 14). Regular screening of structural databases confirmed early results of W. Nowacki [56,57].

Utilizing the Universal Method, Fedorov collected numerous data of crystal morphology and refractive indices. The initial goal of these experiments was to determine the space group or, at least, to identify a crystal from precise macroscopic measurements of face angles. In order to demonstrate the efficiency of his crystallochemical analysis, Fedorov asked Groth to send him some crystals for identification. Groth responded on 22 November 1910 by supplying 28 specimens ([7], p. 140) the nature of which was known to him. In a letter, dated 8 December 1910, Fedorov reported the first results, which Groth confirmed). In 1911 (Manuscript received by the Russian Academy October 26, 1911), he finished the huge data collection "Das Krystallreich. Tabellen zur krystallochemischen Analyse" ("The realm of crystals. Tables for crystallochemical analysis.") describing 10,000 substances on 1050 pages and including an atlas of more than 100 tables [58] (Figure 10). Groth took a lively interest in that to update his textbook "Chemische Kristallographie" (The book, published in four volumes by W. Engelmann (Leipzig), 1906–1919, has been used by Fedorov. When Groth requested updates of interfacial angles presented in his book, Fedorov proposed to send a student to St. Petersburg ([7], p. 145)).

The discovery of X-ray diffraction in 1912 by v. Laue, Friedrich and Knipping [59,60] as well as the first crystal structure analyses by W. L. Bragg and W. H. Bragg [61,62] first reduced the interest in crystal morphology. Moreover, due to World War I, tables could not be printed. Later on, however, the tremendous effort with respect to the wide scope of crystals and the quality of experimental data was appreciated when the Russian Academy published the work and hence preserved this treasure. The author took part in the corrections, but he did not live to see that. The manuscript was sent to the press in 1918. Due to illness and death of Fedorov in 1919 the work appeared not before December 1920.

ЗАПИСКИ РОССІЙСКОЙ АКАДЕМІИ НАУКЪ.

MÉMOIRES DE L'ACADÉMIE DES SCIENCES DE RUSSIE.

VIIIᵉ SÉRIE.

| ПО ФИЗИКО - МАТЕМАТИЧЕСКОМУ ОТДѢЛЕНІЮ. | CLASSE PHYSICO - MATHÉMATIQUE. |
|---|---|
| Томъ XXXVI. | Volume XXXVI. |

# DAS KRYSTALLREICH.

TABELLEN

## ZUR KRYSTALLOCHEMISCHEN ANALYSE.

Von

### E. von Fedorow

unter Mitwirkung von

D. Artemiev, Th. Barker, B. Orelkin und W. Sokolov.

MIT ATLAS.

(Der Akademie vorgelegt am 26. Oktober 1911).

## ATLAS.

ПЕТРОГРАДЪ. 1920. PETROGRAD.

**Figure 10.** E. v. Fedorov, Title page of "Das Krystallreich." ("The realm of crystals.") Petrograd 1920 [58].

In 1912, Fedorov formulated his vision of how to profit from the tables [63], which he expected to appear in the near future. Working out the basics of his crystallochemical analysis, he highlighted the following aspects: (i) Crystallography belongs to the major physico-chemical disciplines. Its main subject is crystal morphology. Numerous researchers have collected data of face angles over the centuries. At the beginning these could successfully be employed to identify a well-crystallized specimen. With increasing number and complexity of known minerals the method became more and more ambiguous. (ii) While the constancy of interfacial angles [64] proved reliable, the crystal forms (Set of all symmetrically equivalent faces) developed under different growth conditions did not, unless Miller indices were referred to standardized coordinates. The latter condition is one demanded by Fedorov from the very beginning of his morphological studies. He also proposed standard designations of crystal forms [32]. (iii) Having collected thousands of crystal forms, Fedorov found that some of them proved stable. Indicating them in the table should be helpful to identify a crystal. On an average not more than 10 forms belonged to a particular substance. Moreover, they were ranked according to their abundance. Furthermore, cleavage planes had been given in order to improve the accuracy of morphological analysis. (iv) The formation of those forms is strongly influenced by admixtures to the solution which provided the growing crystal. In order to provoke the evolution of important forms, growth of spherical crystals from supersaturated solutions had been employed. (v) Given that characteristic forms do exist, they should show up in isomorphic series It was Eilhard Mitscherlich, a German chemist and mineralogist, who observed that some salts of potassium crystallized with the same form [65]. Fedorov identified isomorphic series exhibiting up to 54 members of the formula $(XO_4)_2NM_2.6H_2O$, where X = S, Se, Cr; N = Mg, Mn, Fe, Ni, Co, Cu, Zn, Cd; M = K, Rb, Cs, $NH_4$, Tl [63] (p. 518). (vi) Crystal forms of highest solubility, and hence, of maximum reticular densities

turned out most important. The reticular density is defined as number of 'atoms' per unit area on a specific net plane {*hkl*}. At the turn to the 20th century the concept of 'atom' was still vague. For the evaluation of those densities see [66,67]. Nevertheless, the latter pure geometric entity could serve as first approximation of solubility only. Inclusion of as much as possible additional features was recommended to boost the analysis. (vii) Finally, Fedorov presented numerous examples of real specimens where he explained his crystallochemical procedure.

Although his initial hope to determine the space group did not come true, Fedorov demonstrated successfully the power of quantitative crystal morphology. With the aid of careful measurements of the external shape of a crystal alone it became possible to identify quickly any crystalline substance for which measurements had been recorded in "Das Krystallreich" [58].

### 3.4. 100 Years after Fedorov: Postclassical Crystallography

After the pioneering derivation of 230 space group types by Fedorov, Schoenflies and Barlow (William Barlow's share in the evolution has been dealt with recently [35]) at the end of the 19th century, the concept of symmetry groups has been extended in various manners during the 20th century. Fedorov himself contributed an early step towards symmetry in two dimensions [68] as an amendment to his major publications on the symmetry of finite figures and regular systems of figures [26,69]. He eventually defined a regular system as the entirety of finite figures, which exists in all directions interminable [26] (p. 10) (A supplemented version [70] followed [26]). So, Fedorov derived 17 plane groups or wallpaper groups, which leave a point of the plane fixed when there is no upper and lower side of the two-periodic pattern. Thus, he completed the 13 groups found by Sohncke [71].

Depending on the kind of order revealed with the aid of modern methods of X-ray, neutron, positron or electron diffraction, appropriate concepts of symmetry have been developed. Mentioning only some of them to illustrate the diversity, we refer to the literature for more details.

As an outcome of Fedorov's treatment of two-dimensional patterns, Niggli [72] reactivated work on that topic referring to a growing interest in arts and crafts during the 1920s (Pólya [73] pointed out this demand explicitly). When taking the upper and lower side of a plane into account, 80 plane space groups resulted to describe its symmetry. Several authors (Hermann [74], Weber [75], Alexander and Herrmann [76] and Heesch [77]) focused their work on that. Restraining the dimension further, the symmetry of ornamental border or bordure is concerned, fascinating people for a long time. Seven symmetry groups are retained [78]. In case of a non-polar plane the number of symmetry groups increases to 31 band or frieze groups [79]) (cf. Jones [80]). Eventually, the symmetry of figures characterized by a definite distance to a fixed direction (rod symmetry) had been addressed. For details refer to Hermann [74], Alexander [79], Šubnikov, Kopcik [81].

The first step towards n-dimensional crystallography was accomplished through the admission of four-dimensional point symmetries in three-dimensional space by Heinrich Heesch (* 25 June 1906 Kiel; ✠ 26 July 1995 Hannover; German mathematician [82]) [83] and by Aleksej Vasil'evič Šubnikov (* 29 March 1887 Moscow, ✠ 27 March 1970 Moscow), a Russian crystallographer [84]. Introducing the antisymmetry operation, which changes the character of a given property (e.g., charge changes between plus and minus; magnetic moment between up and down; color between black and white, a.s.o.), the number of point groups increased altogether to 122, i.e., 32 conventional (polar or 'white'), 32 'grew' (containing time inversion as well as rotation/antirotation pairs) and 58 'black-white' groups (Also termed 'Šubnikov antisymmetry classes' [85]). Applying antisymmetry operations to the seven continuous symmetry groups (Curie Groups), which were introduced by P. Curie [86], and worked out further by Schubnikow [87] and Heesch [88], 21 continuous groups resulted, so that the number of black-white point groups increased to 143. For a complete systematics of antisymmetry point groups including continuous groups cf. Šubnikov [89]. The same author described 13 continuous point symmetry groups of the antisymmetry of textures [90].

In 1949 the first structures of magnetic crystals were determined employing neutron diffraction (Shull et al. [91]). It turned out that spin ordering did not always fit the underlying atomic unit cell. Hence the magnetic structure deserved own symmetry concepts. This was the starting point of numerous theoretical and experimental studies, in particular by Šubnikov, Belov and their schools. We just mention a few highlights. When in 1956 Tavger and Zaitsev [92] derived magnetic point symmetries by applying time conversion instead of the antisymmetry operation, they became aware of the identity between their 58 magnetic and Šubnikov's [93] black-white point symmetry groups.

Generalization of Fedorov's space groups (or space group types, see Wondratschek [94]) was accomplished by combining crystallographic symmetry operations with time inversion, anti-translations, anti-rotations, anti-screw operations, anti-reflections and anti-inversions. It was Zamorzaev [95], who derived these groups for the first time. In his thesis (1953), he presented 1652 groups in total. Utilizing a different method, Belov et al. [96] ended up with the correct number of 1651 members. They are termed antisymmetry, dichromatic, black-white, Heesch-Šubnikov or Šubnikov groups. A graphic compilation of their symmetry elements is given by Kopcik [85].

A second direction of evolution shall be mentioned here. One basic column of 19th century crystallography was the postulate of lattice periodicity as indicated at the beginning of the 20th century by sharp spots in the X-ray diffraction patterns. Though various lattice defects at atomic scale had been observed already before World War II, global deviations from periodicity came into focus afterwards only and gave rise to new symmetry concepts. Examples are aperiodic crystals, lacking translation symmetry (Janssen et al. [97]). Among them are incommensurate structures with periodic modulation of the atomic position, which does not fit the lattice periodicity and quasicrystals, the symmetry of which is not compatible with lattice periodicity. A striking feature is the appearance of 'non-crystallographic' symmetries, like fivefold axes (Shechtman et al. [98]). Concepts of description in n-dimensional superspace and in even more generalized space-time enable a wide view on the structure and properties of solids. The reader is referred to [97] for details.

In view of the concept of incommensurate structures, a remarkable example of the efficiency of precise morphological measurements will be recalled. It refers to the mineral Calaverite ($Au_{1-x}Ag_xTe_2$). Smith [99] could not morphologically index all forms unambiguously and supposed an unconventional structure; Fedorov [100] re-evaluated Smith's data, admitted difficulties with indexing, but was afraid of stating a fundamental new order. In 1931, Goldschmidt et al. [101] repeated the morphological measurements, again with anomalous high-indexed faces. It was not before 1989 that Janner and Dam [102] were able to give all crystal faces four low-index indices (hklm) using a fourth basis vector. The conclusion was that calaverite has an incommensurably modulated structure. Consideration of early morphologists proved justified. Recently, the modulation of atom positions has been explained at the level of electron theory [103].

## 4. Conclusions

Fedorov's excellent achievements in the fields of symmetry theory, geometry of polyhedra, microscopy of minerals and rocks as well as crystallochemical analysis constitute fundaments of our present natural science.

Between 1892 and 1915, Fedorov's work became known through the 'Zeitschrift für Krystallographie und Mineralogie' (Z. Kryst.) to a large scientific community in Europe and elsewhere. In this way, he got in contact with German crystallographers, mineralogists, geologists, mathematicians and mechanics, as, for example, with A. Schoenflies, P. Groth, L. Sohncke, C. Klein and R. Fuess.

Intensive correspondence between Fedorov and Schoenflies accelerated the process of validation of 230 space groups. Friendly personal contact as well as comprehensive correspondence with the editor Groth of Z. Kryst. facilitated the creation of precise conceptions and symbols, thus, promoting a high standard of the growing crystallography. As Fedorov was a critical exponent of his field, he was involved in several enlightening discussions in the journal for the benefit of both colleagues and himself.

When browsing through more than 70 articles of E.S. Fedorov written in German, the following features become apparent: (i) A wide field of topics is covered, including both original theoretical and experimental methods. Fedorov developed a four-axis universal goniometer for precise measurement of interfacial angles, the universal table for the microscopic determination of optical properties and various new methods for evaluation of the data obtained. (ii) Unlike numerous other workers in the field of crystallography and mineralogy at that time, Fedorov preferred the application of efficient mathematical methods. (iii) His publications on space groups and structure models derived from them proved essential to pave the way for a quantitative determination of crystal structures utilizing X-ray diffraction by W. L. and W. H. Bragg in 1913.

His extraordinary contribution to crystallography has been acknowledged as an outcome of his active part in the international advancement of crystallography and his particular appreciation by German crystallographers, when he was appointed corresponding member of the Mathematical-scientific Class of the Bavarian Academy of Sciences. The nomination came from Paul v. Groth and was co-signed by Leonhard Sohncke. Shortly before his death he was elected a member of the Soviet Academy of Sciences in 1919.

**Supplementary Materials:** The following are available online at http://www.mdpi.com/2075-163X/10/2/181/s1, Table S1: Publications of E. v. Fedorow in Z. f. Krystallographie and Mineralogie.

**Author Contributions:** Conceptualization, P.P. and S.K.F.; writing—review and editing, P.P. and S.K.F. All authors have read and agreed to the published version of the manuscript

**Funding:** This research was funded by the German Academic Exchange Service (DAAD) and by the Russian Foundation for Basic Research (RFBR, project No. 18-29-12106).

**Acknowledgments:** One author (P.P.) wants to express his gratitude to the organizers of the XIX International Meeting on Crystal Chemistry, X-ray Diffraction and Spectroscopy of Minerals for inviting him. The authors also thank the anonymous reviewers for their constructive comments and useful suggestions.

**Conflicts of Interest:** The authors declare no conflict of interest.

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
