# Peer review of "E. S. Fedorov Promoting the Russian-German Scientific Interrelationship†"

_minerals, doi:10.3390/min10020181_

Round 1
Reviewer 1 Report
The paper “E.S.Fedorov Promoting the Russian-German Scientific Interrelationship” by Peter Paufler and Stanislav Konstantinovič Filatov reconstructs the scientific discovery of E.S. Fedorov in the field of crystallography and his relationship with german crystallographer.
The scientific work of Federov on defining the 230 space groups together with Schoenflies was fundamental to the development of modern crystallography.
The paper is a review of the scientific life of Federov and how he contributed to the development of crystallography. The connection with German mineralogists-crystallographers and the possibility to publish his results not only in russian but in german in the journal “Zeitschrift fuer Krystallographie und Mineralogie” was fundamental to spread his scientific findings in the international scientific community, and not to lose them.
The paper is an historical overview of the role of E.S. Fedorov in the mineralogy and crystallography science. Given the topic of the special issue, I find this paper appropriate and very interesting to read as an introduction to the special issue.
The manuscript is well written but could be shorten in some parts especially if possible in the references list. I also appreciated the par. 4 “100. years after Fedorov. Postclassical crystallography” .
Please note that Fig 2 caption has different font size and seems incomplete.
Some of the note can be embedded in the figure captions like in Fig 3 and note 12.
My suggestion is to publish the manuscript.
Author Response
Point 1: Reviewer 1 proposed a shortening of the list of references. Considering that this review includes little-known papers, however, we hope to provide readers with useful links for further studies when retaining the list.
Point 2: Following Reviewer 1, caption 2 has been completed by adding '(Photo: S.K.Filatov)'.
Point 3: As proposed by Reviewer 1, note 12 was changed accordingly:
"The symmetry of infinite regular systems of figures". Unlike this announcement, the final version is displayed in Figure 3.
Reviewer 2 Report
I would like to thank the Editor for the opportunity to read and evaluate this extremely interesting manuscript. I learnt much new, even though I am rather deeply involved in crystallographic research and am interested in the history of this field. I am confident that the readers will learn much, and this knowledge will be both exciting and timely. The paper pays proper tribute to the founders of crystallography, and it also teaches many lessons of how research collaboration can and should be conducted. The paper is therefore not of merely "historic value", but is also important to teach younger generations the principles of the scientific ethics. In this respect, it is also valuable that the paper on the history of Russian-German collaboration is co-authored by two world-recognized crystallographers representing the two nations deeply involved in the foundations of crystallography, namely a Russian and a German. This helps to link the present time with the events that happened a century ago.
I support enthusiastically the publication of this paper and look forward seeing it published.
I have a few minor comments.
The first comment is on a spelling issue. Apparently, spelling of Crystallography in German changed in the last 100 years. Now it is Kristallographie, not Krystallographie (https://link.springer.com/journal/12379), i.e. not
Zeitschrift für Krystallographie und Mineralogie, but
Zeitschrift für Kristallographie und Mineralogie.
When old references are given, the old spelling looks appropriate. However, possibly some comment at the very beginning of the paper could help the readers to avoid some uncomfortable feeling of a misspelling. Also consistency in the two types of spelling is necessary. For example, at line 239 we see: “Das Krystallreich. Tabellen zur kristallochemischen Analyse” - two different spellings in the same book title.
I also see some discrepancy in the spelling of a name Jelij / El'evic:
Jelij Evgrafovič Fedorov (-)
Vadim El’evič Cubin
Either Elij / El'evic, or Jelij / Jel'evic
Also what (-) should mean instead of the dates of birth and death?
69 years after finishing hisstudies - should be:
69 years after finishing his studies
Thätigkeit - should not this word be spelled Tätigkeit?
54 Aleksej Vasil’evič Šubnikov (*29.03.1887 Moscow, U 27.0,3.1970 Moscow) Russian crystallographer.
Should be:
54 Aleksej Vasil’evič Šubnikov (*29.03.1887 Moscow, U 27.03.1970 Moscow) Russian crystallographer.
379 Tom 2. Izd-vo ANSSSR: Moskwa 1951, 297-313.
Should be:
379 Tom 2. Izd-vo AN SSSR: Moskwa 1951, 297-313.
Since most references are either in German, or in Russian, it may be a good idea to accompany all the titles by English translations. This has been done for some of the references, but not for all of them.
372-373 Russian Fonds of Fundamental Investigations (RFFI)
I am more familiar with the English abbreviation RFBR - Russian Foundation for Basic Research, please double-check
232-235 Certain combinations of space symmetry are obviously preferred for molecular packing. Whereas the most frequently observed space group of minerals and inorganic compounds is the orthorhombic Pnma (No.62, see white background of Graphical Abstract), the tetragonal space group P42cm (No.101, black background) so far seems to be avoided completely by any mineral (Filatov [54], see also Graphical Abstract).
This statement is interesting, but I would suggest that it should be slightly revised and extended.
First, it is not appropriate to speak about "molecular packing" when discussing minerals and inorganic compounds, in which no individual molecules are present in the crystals. This must be re-phrased.
Second, if the preference of "molecular packing" for certain combinations of space symmetry elements is discussed, one must also mention the population of the symmetry groups of molecular structures, which is different from that for inorganic compounds and minerals, P21/c and C2/c being the most populated space symmetry groups. The work by Nowacki and Kitaigoroskii deserves being mentioned here:
Nowacki, W. (1942), Helv. Chim. Acta, 25, 863-878; Nowacki, W. (1943), Helv. Chim. Acta, 26, 459-462
Kitaigorodski, A. I. (1962). Organic Chemical Crystallography, Consultants Bureau, New York, 1961; AI Kitaigorodoski and KU Mirskaya. Soviet. Phys. Cryst, 6, 408.
327 even more generalized four-dimensional space-time
Since incommensurately modulated phases and quazicrystals - what is not the same - can require more than 3 indices to describe space structures, a generalized space-time description is more than four-dimensional. I would delete "four-dimensional" in this context, and just leave "even more generalized space-time"
148-150 Nowadays, it seems unreal, to find editors taking personal interest in both the science and personality of authors on the one hand and authors contacting editors on a personal level on the other
I would suggest to soften this statement and not make it absolutely general. I myself was fortunate in my life to contact many editors "taking personal interest in both the science and personality of authors on the one hand and authors contacting editors on a personal level on the other". I agree that this may be not always the case, but this is no way "unreal".
358-360 (ii) Unlike numerous other workers in the field of crystallography and mineralogy, Fedorov preferred the application of efficient mathematical methods.
I would not make this statement so general. To the best of my knowledge, "numerous workers in the field of crystallography and mineralogy", contrary to this statement, "apply efficient mathematical methods". Maybe the authors aimed to express something else? Try to re-phrase.
Author Response
Point 1: Reviewer 2 proposed to comment on the different spelling of 'Krystallographie/Kristallographie' in the manuscript at the beginning of the article.
The additional note No.3 has been inserted :
"When quoting German words, the spelling of the original document is used. Hence, ‘crystallography’ is spelled in German ‘Krystallographie’ before the German spelling reform in 1921 and ‘Kristallographie’ afterwards. Readers accustomed to modern German should be aware of that, because it applies to some more terms in the text or in the references. "
Point 2: Line 262 _Inconsistency of spelling 'Krystallreich' and 'kristallochemisch' .
This misprint has been corrected: "Das Krystallreich. Tabellen zur krystallochemischen Analyse."
Point 3: Uniform spelling of Jelij /El'evic required.
Has been corrected. Uniform transliteration of the cyrillic letter "Э" by "Ė" was done.
Point 4: Also what (-) should mean instead of the dates of birth and death?
Note 7 inserted: "Data of birth and death not available."
Point 5: 50 years after finishing hisstudies - should be:
50 years after finishing his studies.
Done.
Point 6: Thätigkeit - should not this word be spelled Tätigkeit?
At this place --> No. (see Note 3 and Point 1 )
Point 7:
56 Aleksej Vasil’evič Šubnikov (*29.03.1887 Moscow, U 27.0,3.1970 Moscow) Russian crystallographer.
Should be:
56 Aleksej Vasil’evič Šubnikov (*29.03.1887 Moscow, U 27.03.1970 Moscow) Russian crystallographer.
Done.
Point 8:
405 Tom 2. Izd-vo ANSSSR: Moskwa 1951, 297-313.
Should be:
405 Tom 2. Izd-vo AN SSSR: Moskwa 1951, 297-313.
Done.
Point 9:
Since most references are either in German, or in Russian, it may be a good idea to accompany all the titles by English translations. This has been done for some of the references, but not for all of them.
Point 10:
372-373 Russian Fonds of Fundamental Investigations (RFFI)
I am more familiar with the English abbreviation RFBR - Russian Foundation for Basic Research, please double-check.
Changed to:
397 Russian Foundation for Basic Research (RFBR, project No. 18-29-12106).
Point 11:
First, it is not appropriate to speak about "molecular packing" when discussing minerals and inorganic compounds, in which no individual molecules are present in the crystals. This must be re-phrased.
Second, if the preference of "molecular packing" for certain combinations of space symmetry elements is discussed, one must also mention the population of the symmetry groups of molecular structures, which is different from that for inorganic compounds and minerals, P21/c and C2/c being the most populated space symmetry groups.
Re-phrased version:
253 Certain combinations of space symmetry are obviously preferred for the packing of given structural units.
257 The distribution depends, however, on the symmetry of the structural unit. Organic compounds, for instance, occur most frequently with a crystal structure of the monoclinic space group P21/c.
Additional note 39:
Regular screening of structural databases confirm early results of W. Nowacki (Helv. Chim. Act. 1942, 25, 863).
Point 12:
Since incommensurately modulated phases and quazicrystals - what is not the same - can require more than 3 indices to describe space structures, a generalized space-time description is more than four-dimensional. I would delete "four-dimensional" in this context, and just leave "even more generalized space-time"
353 now reads:"even more generalized space-time"
Point 13:
148-150 Nowadays, it seems unreal, to find editors taking personal interest in both the science and personality of authors on the one hand and authors contacting editors on a personal level on the other.
162 "Nowadays, it is rarely to find editors"
Point 14:
384 (ii) Unlike numerous other workers in the field of crystallography and mineralogy, Fedorov preferred the application of efficient mathematical methods.
384 Now reads: Unlike numerous other workers in the field of crystallography and mineralogy at that time, Fedorov preferred the application of efficient mathematical methods.
Reviewer 3 Report
Paufler and Filatov give an invaluable insight into Fedorov’s scientific contribution into the development of space group theory and its legacy. The manuscript is well written and requires just minor polishing step before the publication, the authors should do the proof-reading. For example, line 46 no years are given for Jelij, line 71 no credit for the photo is given, line 103 four full stops instead of 3, etc
Author Response
For example, line 46 no years are given for Jelij, line 71 no credit for the photo is given. For example, line 46 no years are given for Jelij, line 71 no credit for the photo is given, line 103 four full stops instead of 3, etc, line 103 four full stops instead of 3, etc
Done